# Considering Health Literacy, Health Decision Making, and Health Communication in the Social Networks of Vulnerable New Mothers in Hawai‘i: A Pilot Feasibility Study

**DOI:** 10.3390/ijerph17072356

**Published:** 2020-03-31

**Authors:** Tetine Sentell, Joy Agner, Ruth Pitt, James Davis, Mary Guo, Elizabeth McFarlane

**Affiliations:** 1Office of Public Health Studies, University of Hawai‘i at Mānoa, Honolulu, HI 96822, USA; ruthpitt@hawaii.edu (R.P.); ecmcfarl@hawaii.edu (E.M.); 2Community and Cultural Psychology Department, University of Hawai‘i at Mānoa, Honolulu, HI 96822, USA; joyagner@hawaii.edu; 3John A. Burns School of Medicine, University of Hawai‘i, Honolulu, HI 96813, USA; jamesdav@hawaii.edu; 4School of Nursing and Dental Hygiene, University of Hawai‘i School of Nursing and Dental Hygiene, Honolulu, HI 96822, USA; maryguo@hawaii.edu

**Keywords:** health literacy, social networks, health communication, native Hawaiian mothers, Filipino mothers, low-income mothers

## Abstract

Health literacy is understudied in the context of social networks. Our pilot study goal was to consider this research gap among vulnerable, low-income mothers of minority ethnic background in the state of Hawai‘i, USA. Recruitment followed a modified snowball sampling approach. First, we identified and interviewed seven mothers (“egos”) in a state-sponsored home visiting program. We then sought to interview individuals whom each mother said was part of her health decision-making network (“first-level alters”) and all individuals whom the first-level alters said were part of their health decision-making networks (“second-level alters”). Health literacy was self-reported using a validated item. A total of 18 people were interviewed, including all mothers (n = 7), 35% of the first-level alters (n = 7/20), and 36% of the second-level alters (n = 4/11). On average, the mothers made health decisions with 2.9 people (range: 1-6); partners/spouses and mothers/mothers-in-law were most common. One mother had low health literacy; her two first-level alters also had low health literacy. Across the full sample, the average number of people in individuals’ health decision networks was 2.5 (range: 0–7); 39% of those interviewed had low health literacy. This can inform the design of future studies and successful interventions to improve health literacy.

## 1. Introduction

### 1.1. Background

While health literacy is often defined as “the degree to which individual*s* have the capacity to obtain, process, and understand basic health information and services needed to make appropriate health decisions,” [1] few individuals experience chronic illness, confront health crises, and/or make health decisions entirely alone [2,3,4,5,6,7,8]. For this reason, scholars have called for more empirical research on socioecological influences in health literacy [2,3,6,7,9,10,11,12]. Conceptualizing health literacy within social networks can capture how skills and knowledge are leveraged for health-related decisions and actions, and how they can be utilized to strengthen health interventions [13].

This perspective may be particularly relevant for communities, including many minority and ethnic communities, with a collective and/or family sociocultural orientation [14,15,16,17]. Research considering health literacy in a social context is important for creating and sustaining meaningful interventions to improve health for individuals, families, and communities, especially towards achieving health equity. Such initiatives are responsive to existing sources of community strength and resilience as well as preferences [16].

Social network analysis (SNA) considers social structures constructed through interaction [18,19,20]. A social network perspective on health literacy can effectively represent our experiences in which health and health behaviors are strongly impacted by relationships and social context [21,22]. SNA provides a useful methodological tool to help fill the socioecological research gaps in health literacy as it has the potential to address both the characteristics of individuals within networks, and the characteristics of the network as a whole [2,23,24]. SNA can also be used to research the spread of health literacy through a network over time [25]. 

Social networks powerfully impact health [24,26]. For instance, having a close friend who smokes predicts smoking [27]. Recent innovative studies have leveraged social networks to change health behaviors, improve health knowledge and health-related decision making, and to increase the effectiveness of health behavior interventions [13]. Strong and active social networks may be particularly important for vulnerable communities with health disparities [16]. 

Health literacy is also a powerful predictor of health. A large body of literature has found that higher health literacy at the individual level is associated with better health outcomes, better health care access, and more positive health-related behaviors [28,29,30]. The reasons why individual health literacy is such a strong health indicator are not fully understood [31]. Improving the consideration of the effect of social networks in health information gathering, processing, and decision making may help strengthen a weakness of existing health literacy research. 

A small and growing body of evidence supports the relevance of social networks in health literacy specifically. Several lines of research suggest community, social network, and/or family-level interventions to improve health literacy can effectively improve health outcomes. Both community-level health literacy and individual-level health literacy have been independently associated with poor self-reported health [10]. Social networks are expected to be particularly relevant to those with low health literacy. Social capital has been found to attenuate the effect of low health literacy on health information self-efficacy and information-seeking intentions [32]. 

Qualitative studies have found that individuals with low health literacy specifically draw upon social networks to better understand health care concerns [33]. Communities with a lower level of health literacy may place a “greater reliance on personal experience and information obtained through lay networks” ([34], p. 867). Qualitative methods have found that “distributed” health literacy plays an important role in health outcomes [2,4,5,35,36]. SNA methods may provide critical insights on these topics from a quantitative perspective [2]. 

### 1.2. Study Focus

This pilot study used an egocentric SNA method to consider health literacy. Egocentric analyses focus on an individual (termed the “ego”) describing their own network members (termed “alters”). Our egos were mothers, as both health literacy and social networks are particularly relevant for parents [4]. Parents with lower health literacy have less knowledge about health outcomes, less healthy behaviors, and less optimal use of health services [28,30]. Lower health literacy is associated with important health outcomes in low-income mothers, including less keeping of prenatal appointments, more smoking, and less breast feeding [22,29,36]. Children of parents have poorer health literacy have poorer health outcomes than children of parents with higher health literacy [28,30]. Promisingly, studies have found health literacy interventions in parents can have a positive impact, potentially increasing health-related knowledge and behaviors overall [28,30], suggesting improving health literacy in a network can show significant health-related returns. 

As the social network perspective is particularly relevant for improving health equity, we focused our study on those with known health disparities. Across most communities, socioeconomically vulnerable mothers have unmet health needs [37]. Socioeconomic vulnerability is influenced by a combination of risk factors at both the individual level and environmental level [38]. Both individual and community poverty are particularly important, impacting access to factors such as housing, food, transportation, clothing, health care, and education that have both immediate and future consequence to life course trajectories [38,39]. Socioeconomically vulnerable mothers are ideal to consider in network health literacy as they have known health literacy challenges [40], and often tend to their own health needs as well as those of their children, parents, and others [41]. Intervention efforts with socioeconomically vulnerable mothers can not only impact child well-being, but can also have ripple effects in vulnerable families and communities.

We also focused on two racial/ethnic groups that have health disparities within our specific context, the state of Hawai‘i. We include Native Hawaiians and Filipinos due to the cultural relevance of the collective in Pacific Islander communities [15,42,43] as well as the importance of improving health in these populations to achieve health equity. Native Hawaiians and Filipinos are also some of the fastest growing racial/ethnic population in the United States (US) and are understudied generally [44]. They make up two of the five major racial/ethnic groups in Hawai‘i, the study location, comprising an estimated 38.5% (21.3% Native Hawaiian and 17.2% Filipino) of the state’s total population [45]. Compared to the other major racial/ethnic groups in Hawai‘i, Native Hawaiians and Filipinos often have poorer health outcomes, including high rates of chronic conditions and maternal and child health inequities, as well as significant socioeconomic challenges and limited access to high quality, culturally relevant health care [46,47,48,49,50]. The consequences of these inequities include an over 10-year gap in life expectancy between Native Hawaiians and Chinese, the longest lived racial/ethnic group in the state, and a 6-year gap between Filipinos and Chinese [45]. Relevant to health literacy, Filipinos and Native Hawaiians are less likely to have college degrees than other major racial/ethnic groups in the state and are disproportionately represented among lower-income households. Relevant to the network approach, Native Hawaiian and Filipino family and household sizes are often larger than seen in the general US population.

### 1.3. Methodological Considerations 

Because using SNA methods to describe the health literacy context of low-income Native Hawaiian and Filipino mothers was a new method, there were some complexities to consider. First, while some formative SNA looked at networks 3 levels out [27,51], for practical reasons, much egocentric SNA research looks only at one level of alters. Given our focus on health literacy and collectivist cultures, we felt one level of alter might not be sufficient to fully understand the broader network context in which health information flows and to best see how health literacy might be achieved and sustained by low-income mothers. Thus, our study sought to identify and describe not only the network that an individual mother uses to talk about health issues, but also networks of the mothers’ alters.

Additionally, in egocentric SNA methods, the ego is typically asked to describe the characteristics of network alters. Yet to ask the ego to disclosure alters’ health literacy was also problematic. Research shows health literacy may not be readily known to others, including friends, family, and providers. In many cases, those with low health literacy may feel ashamed and not tell even close family members if they have trouble understanding health information or systems [52]. Further, while the level of educational attainment might be known to the ego, this cannot proxy health literacy, which can be both overestimated and underestimated by educational attainment [53,54]. As we wanted to map the health literacy capacity of the network, not only of the individual mother, we directly assessed alters’ health literacy. 

Finally, social network studies among low-income communities can present unique challenges. Low-income mothers are often juggling multiple simultaneous demands, making recruitment a challenge. They may also be harder to reach via phone or text-based methods. Low-income populations can have unique ways of using cell phones, such as sharing phones, changing cell phone numbers often, and receiving many text messages a day. This may translate to participants not receiving text messages, having special privacy concerns, higher drop-off rates, or less attention to text-based data collection methods [36]. Furthermore, they may not have a desire or time to participate in research, especially when trust is not clearly established.

### 1.4. Specific Study Goals 

Given these challenges, our study had two goals. The first goal was to assess feasibility. We wanted to determine whether it was feasible to interview (a) vulnerable mothers; (b) individuals whom the mothers included in their health decision making (first-level alters); and (c) individuals whom the first-level alters included in their health decision making (second-level alters). Feasibility findings, including what percent of alters were reached and what strategies were successful or unsuccessful, can inform future studies on this topic. The second goal was to describe the mothers’ health literacy in the context of their social networks. We examined network size (average number of discussion partners and range); concordance in the mothers’ decision-making networks (whether alters and egos have similar health literacy and education); the composition of decision-making networks (family versus friends); and rationale (why they chose to include those people specifically). This pilot feasibility study was designed to both test methods and provide preliminary data from a small sample that can inform the research field around this important topic. This paper also provides landscaping around methods and considerations on the social network approach generally. 

## 2. Materials and Methods 

### 2.1. Sampling

Recruitment had three steps and followed a modified snowball sampling approach. Snowball sampling is commonly used in egocentric social network research because it is a logical approach to identifying contacts within the ego’s network [55]. It is also presents advantages in accessing hard-to-reach populations that may have less trust in researchers because the researcher is connected to a known source [56]. We modified the snowball approach by identifying the seven initial focal mothers first through the home visiting program as described in more detail below. 

We performed this work in Honolulu, Hawai‘i, as this state is extremely diverse racially, particularly for Asian American (Filipino) and Pacific Islander (Native Hawaiian) samples, making it an excellent location to consider these understudied racial/ethnic groups. This study was approved the UH Human Subjects Institutional Review Board (Protocol #23485).

First, we started with 7 primary, focal individuals referred to as “egos.” The ego inclusion criteria were that the women had at least one child, self-identify as Native Hawaiian or Filipino, have low income (family income <200% of the federal poverty level), be between 25 and 55 years old, and be eligible for participation in a voluntary, evidenced-based home visiting program [57]. This program supports socioeconomically vulnerable mothers with the aims of promoting maternal and child health, fostering positive parenting and children’s school readiness, and preventing child abuse and neglect [57]. To be eligible for the home visiting program, mothers have to have low income and live in communities with a concentration of vulnerabilities that include higher rates of unemployment, poverty, high school dropout, child maltreatment, substance abuse, interpersonal violence, crime, low birth weight, premature birth, and maternal mortality as compared to other communities in the state [58]. Each mother provided informed consent, was interviewed in person, and received a $30 incentive. 

Second, from each of these interviews, we identified who each of the mothers would include in important decisions around their health (first-level alters). Specifically, they were asked “Imagine you had to make a decision about your health. For instance, you needed to decide whether to take a medication that had significant side effects or needed to decide if you needed a second opinion about a diagnosis. Who would you talk to about this health care decision?” We then interviewed first-level alters following the same methods we used with the mothers. These individuals also receive $30 incentives.

Third, we asked the first-level alters the same question, “Imagine you had to make a decision about your health…” to identify second-level alters. We contacted the second-level alters and interviewed them with a shorter interview protocol. The final set of interviews was designed to be conducted by telephone or online. These individuals received $25 (as their interviews were shorter). 

There were no age, gender, race/ethnicity exclusion criteria for the first- or second-level alters. Inclusion criteria included speaking English, being at least 21 years of age, and the ability to provide informed consent. If the alter was present with the ego, they were invited to participate at the initial interview or at another time. For other alters, following best practices, we attempted three texts or calls per person to reach them for an interview from the contact information provided by the ego or first-level alter. 

### 2.2. Measures

Interviews were semi-structured and took approximately 30 minutes for the egos and first-level alters. Interviews for the second-level alters were typically less than 20 minutes. All were conducted between June and November 2016. In all interviews, we measured self-reported health literacy using the well-validated Single Item Health Literacy Screener (SILS) [57]. In the SILS, participants were asked “How confident are you filling out medical forms by yourself?” A response of “Somewhat,” “A little bit,” “Not at all” was coded as low health literacy. A response of “Extremely” or “Quite a bit” was coded as adequate/high health literacy. In all interviews, we also measured (1) basic demographics, including age, sex, education; (2) who they turn to for health questions and/or decisions; and (3) who turns to them for health questions and/or decisions. In all focal interviews, we additionally explored the participants’ perceptions of how their social network affects their ability to access, interpret, and act on health information using open-ended questions.

### 2.3. Analyses

Given the focus of the study on feasibility and network characteristics, analyses were primarily descriptive. For feasibility, we considered the number of people described by egos and first-level alters, the number of alters who agreed to interview, and whether their self-reported health literacy was concordant with education. For mapping health literacy in context, we considered the networks from the perspective of each mother and all the alters mentioned by her and her network members. This essentially creates a multi-level egocentric study, where the first- and second-level alters are asked the egocentric questions. Everyone nominated by the mother, or the first- and second-level alters is considered part of the larger network that originated with the focal mother, but this is not a full sociometric network structure. From each egocentric decision-making network, we considered the average number of alters, the composition of their networks (family, friend), why they reached out to that person to discuss health matters, and concordance (homophily) in health literacy, age, sex, ethnicity, and education.

## 3. Results

### 3.1. Overview of the Seven Networks

Forty-seven distinct individuals were identified, including seven mothers, 20 first-level alters, 11 second-level alters, and nine third-level alters. Five additional individuals identified by alters as part of their networks were already in the sample as egos or alters. The entire networks of the seven mothers (A-F) are visualized in Figure 1. 

Those we were able to interview are filled in based on self-reported health literacy. Low health literacy is indicated by pink fill. Adequate/high is indicated by grey fill. Those we tried, but could not interview, are outlined in red. Those who were not to be interviewed according to the study protocol (third-level alters) are outlined in blue. Gender is noted in the description. Because this figure contains multiple layers of information about the overall networks, the results are parsed and outlined below according to our original aims.

Some interesting findings are shown in Figure 1 regarding the size and composition of the mothers’ networks, as well as concordance in health literacy and education, and who we were and were not able to interview. Regarding network composition, partners/spouses and mothers/mothers-in-law were most commonly consulted for health information. Among the five males interviewed, all were husbands or boyfriends, and four described reciprocal relationships with the focal moms, meaning they also relied on them as health discussion partners. Two male partners’ health decision networks included only their wives (Mother A and Mother B); another’s included only his wife (Mother C) and his uncle.

Some individuals, both mothers and alters, had very small networks. Mothers A, D, and F reported only two individuals in their health decision network. For instance, Mother F had her husband and her mother, and Mother C reported only one individual in her health decision network (her partner). Several respondents, all male, were “isolates,” talking to no one in their health decisions. For instance, Mother F’s husband did not report anyone in his health decision network, though his wife reported him in hers. Others, however, had large networks, including Mother B and one second-level alter (in Mother A’s network). Overall, only two respondents reported a health information network of four or more individuals.

### 3.2. Feasibility

According to our protocols (interviewing mothers, first- and second-level alters only), we would have liked to interview 38 of the 47 individuals identified. Eighteen consented to be interviewed, which included all mothers (n = 7), 35% of the first-level alters (n = 7/20), and 36% of the identified second-level alters (n = 4/11). Because gender and education can be provided by proxy, we were able to attain that information for everyone in the network, including people who we were not able to interview.

These data have the potential to inform the feasibility of this work going forward and identify any differences that might make recruitment a challenge. Individuals who were identified but were not interviewed were less likely to have completed high school and were more likely to be female. The gender finding may be explained by the fact that males in the study were also more likely to be romantic partners (see Figure 1), and therefore more likely to have been present during the initial interview to recruit or familiar with the study from their partner. 

### 3.3. Characteristics of Interviewed Participants

As shown in Table 1, all mothers had completed high school, but none had gone to college. The alters interviewed were 54.6% female. One alter had not completed high school. Eight graduated from high school. Two had a college degree or higher. Table 2 provides more detailed information about race, age, and health literacy among the 18 individuals who were interviewed. The majority of the mothers self-reported Native Hawaiian (4) as their primary race/ethnicity, and the alters primarily self-reported Native Hawaiian (3) or Filipino (3) as their primary race. Average age among the mothers was 27 years, and among alters was 30 years. Only one mother (A) had low health literacy; her two first-level alters also had low health literacy. The second-level alters did not.

### 3.4. Network Size 

For mothers, the average number of people in their health decision-making networks was 2.9 (range: 1–6). Across the full sample, the average number of individuals in health decision-making networks was 2.5 (range: 0–6).

In general, men had fewer health discussion partners in their networks. Considering just the number of alters of the five male network members who participated in this study, the average number of people talked to was 1 (SD 0.71), with a range of 0–7. This was in comparison to the (13) women who had an average of 2.92 (SD: 1.98). A t-test with no expectation of direction found this difference to have marginal significance (*p* = 0.0530).

### 3.5. Rationale for Choosing Health Discussion Partners

We also considered why individuals included others in their health decisions and identified themes around these findings. The most common reason was due to feelings of closeness and family/family-like ties. Trust was an important determinant. One respondent described: “*I wouldn’t see myself talking to a stranger about my health.”* Another described *“He’s my best friend and I trust him with my life.”* Trust and closeness were underpinned by experience. Relevant to health literacy in social networks, a number of respondents mentioned how they turn to people for advice, especially those with expertise. For instance, one noted she turned to her mother-in-law because *“She’s a lot older so she has experience, knows where to get info.”* Another mentioned about her mother: *“Because she has been through it….She raised 13 kids and some of them have asthma.”*


Other reasons included the practicalities of keeping important people informed. One mom described why she talks to her husband about her health and the health of their children: “*I feel it is important for him to know how I am doing and how they are doing.”* Respondents also talked about how others were part of their decision-making networks because they participated in their health care. For instance, one noted: “*Because she is my wife. She drives me to my appointments.”* Others noted the importance of emotional support and caring from those who help them make health decisions. When asked why they included individuals in their health decisions they described, *“Because she cares very much”* and *“She cares for my well-being.”*

Although less common, some respondents also noted that others in their social network are not always involved in a helpful way. One noted that she turns to her mother *“only when need family history.”*

### 3.6. Network Concordance by Health Literacy and Education

Thirty-eight percent of those interviewed had low health literacy—14% of the egos and 55% of the alters. For the 11 first- and second-level alters that were interviewed, five (45.4%) were concordant with the person who had named them in terms of health literacy (see Figure 1). One mother had low health literacy; her two first-level alters also had low health literacy.

Education data were available by proxy for the entire network, and so concordance could be assessed for more relationships. Twenty-five of the relationships (59.5%) (see Figure 1) were concordant between alters and the people who had named them in terms of education. This indicates that, within this sample, education concordance was more prevalent than health literacy concordance. Similar to past research, among the 18 individuals interviewed, a variety of self-reported health literacy levels were seen across education levels, indicating the importance of measuring health literacy of each individual. 

## 4. Discussion

This study found that taking a social network perspective was relevant to the health communication and health decision experiences in vulnerable mothers. Specifically, our findings provide important evidence on the feasibility of completing this type of study, as well as the size and composition of health information networks of vulnerable mothers. Vulnerable mothers turned to others for consultations on health decisions for a variety of important reasons. However, these networks were often small and constrained to family. Previous research on composition suggests that the number of key members in an individual’s decision-making network is generally at least four [59,60]. The focal mothers in our study did not report four individuals on average in their health information networks. These smaller networks may indicate that the focal mothers are more isolated. Smaller networks may provide less access to information or critical appraisal skills which are necessary for informed decision making. If no one in your network can teach you how to navigate health care encounters or make health decisions, you may turn to the media, which may vary in information quality, or unreliable sources for health information [61]. 

The small networks of the focal mothers may not seem surprising given their socioeconomic vulnerability qualified them for the home visiting program in the first place. However, the relationship between social network characteristics and income is complex. People of lower income tend to spend more time socializing with their network contacts, particularly neighbors [62], suggesting they do not necessarily have smaller networks. However, lower income groups and racial minorities also report less access to “experts” who may be particularly useful network members [63]. Such factors are important to consider in measurement and discussion around vulnerability, health communication, and community resources. Helping to support and create community-based peer experts, through mechanisms such as community health worker programs or support groups may be a critical way to build relevant health expertise into naturally occurring, meaningful social networks.

Our study provides new evidence concerning the interrelations between vulnerability, health communication, and health decision making of ethnic minorities generally and specifically related to Native Hawaiian and Filipino populations. We focused on socioeconomic vulnerability in designing our sample of mothers. We found that these respondents had small health decision networks, while self-reporting racial/ethnic backgrounds often associated with a collectivist orientation. This may be considered another type of vulnerability [64]. While “vulnerability” is a complicated issue with many causal, interrelated factors, another hallmark may be the “inadequacy of interpersonal networks and supports” [64]. At the same time, social networks and social capital can even ameliorate disparities traditionally associated with income and education gaps [65]. For example, Choi found while Filipino immigrants in Hawai‘i have lower income and educational attainment than Korean immigrants in Hawai‘i, access to high levels of health care resources and social capital within the Filipino community provided Filipinos significantly better health care access than Koreans [15]. This supports the idea that social network focused health literacy research and interventions can be utilized to address health disparities among for low-income communities. Community participation in health care activities and decisions are valued in many populations with substantial and persistent health disparities, including the communities studied here. 

Social resources are a critical coping mechanism for many racial/ethnic minority communities [14,18,23]. Considering interventions that support the strength of strong social networks and connectivity in racial/ethnic minority communities may help to promote positive behavior change, better patient education, and shared decision making in diverse communities, providing strength across vulnerabilities [66]. Storytelling interventions have been found to be meaningful, effective, and culturally relevant in Native Hawaiian populations in Hawai‘i [67]. Narrative interventions have also been found to be effective in promoting conversations and shared decision making across diverse populations to support patient preferences in Hawai‘i [68]. Tailoring health communication interventions to support racial/ethnic minorities and leveraging social networks and other community preferences and resources are important areas for future research to support health equity in Hawai‘i and elsewhere. 

### 4.1. Strategies 

A key study goal was to assess the feasibility of interviews for future health literacy study to inform future studies on the topic, including strategies that were successful or unsuccessful. We wanted to determine whether it was feasible to interview (a) vulnerable mothers, (b) individuals whom the mothers included in their health decision making (first-level alters); and (c) individuals whom the first-level alters included in their health decision making (second-level alters).

We found that it was feasible to interview the mothers, especially through the sampling process of a connection from the home visiting program. Alters were a greater challenge to recruit with our planned strategies (using the name/phone numbers given to us by the ego to contact the alter to participate; using three attempts to contact the alters). These gaps significantly limited our ability to understand the full social network context of our focal mothers. Specifically, we were able to interview 18 out of 38 study participants whom we would have liked to interview for a total of 47.4% success. However, if we had interviewed more first- and second-level alters, the networks would have certainly been larger. Thus, we missed the full portrait of the networks.

This could have been because alters were not engaged in our research through a trusted liaison (unlike the mothers connected through their home visiting program). This suggests that engaging willing egos to recruit alters could be a useful strategy by creating a trusted connection, such as calling or texting to try to set up the alter’s interview during the ego’s interview when the ego can explain the situation. Further, incentivizing egos specifically when alters participate (e.g., $5 for each completed interview in one’s network) may be a method towards more complete data collection in future work. Without this information, we could not capture the size and characteristics of the network composition around the new mothers.

Once we were able to interview focal moms and alters, the interview was acceptable to participants and they were willing to describe their networks and answer questions. We learned the sample sizes of networks, finding that the size of networks was lower than we had expected. This would allow us to develop realistic budgets for future research projects (e.g., $15 gift card for the interview participation; $5 gift cards upon completion of alters’ interview completion for egos; an expectation of networks of 2–3 individuals). Before this study, we did not have a good sense of how large a sample would result from a certain number of focal mothers. We also know that the size of networks likely varies by gender. These data will be important to collect for future research feasibility and power calculations for future study proposals. 

Given the goals of the home visiting program, we were also surprised no participant mentioned their home visitor and would add an additional clarifying question about this in the future to understand whether this is because they did not see the home visitor as a resource for personal health decision making, considered the relationship as outside their social network, or for other reasons as yet undetermined. 

Further, while we sampled a vulnerable community, only one of the focal mothers had low health literacy. This made it impossible to compare networks by health literacy from the egocentric perspective of the mothers. A wider sample would have more variety in health literacy, but there are trade-offs between size and feasibility in the depth of the network sample. Future work will need to consider whether it would make sense to sample on education (e.g., x with less than a high school degree, x with a high school degree, x with a college degree) or even health literacy. Education is more feasible, as these data are more likely to be available in advance to plan. However, education does not proxy health literacy as discussed earlier and as shown in Table 3. In Hawai‘i, as elsewhere, previous work has found that low health literacy varies by race/ethnicity. For instance, a previous study in Hawai‘i found that low health literacy varied significantly from 23.9% among Filipinos to 15.9% in Native Hawaiians to 13.2% in Whites [69]. This would impact the expectation of who would need to be screened from a general population sample to get an even distribution by health literacy in groups of interest.

### 4.2. Considerations 

Past social network research has found that homophily may impact social network effects on health because it can influence the spread of ideas and behaviors in social networks [24]. Homophily and bonding social capital are based on homogeneous networks that have similar demographic characteristics such as age, ethnicity, income, and education [70]. The value of bonding social capital lies in the provision of emotional, evaluative, and instrumental supports. In our sample, homophily in terms of gender and age was not particularly important, but egos and alters were often concordant in terms of education. We did not assess homophily in terms of income or ethnicity. Our findings on why mothers chose certain individuals as health discussion partners show that proximity, closeness, and trust mattered. Mothers included people in their health care decision making whom they found caring, who had been through the experience before, or with whom they had long and sustained relationships. Often, they described individuals providing instrumental supports as well, such as taking them to doctor’s appointments or providing childcare. Based on past research, these relationships likely affect mothers’ health directly through psychological mechanisms [71] such as enhancing personal control [72,73], and by increasing self-efficacy for finding, understanding, and using health information [32].

It is important to note that it is not always positive to include others in your health decisions, and social networks are not always health promoting [51,74]. They may instead be forces for exclusion and provincialism as well as sources of stress [32,75,76]. Past research with people with mental illness shows health discussion partners enhanced their health overall, but the effect was significantly reduced when health discussion partners had a negative view of health care or the medical system [77]. Furthermore, misinformation or maladaptive health views can spread through social networks. For example, a large sociometric study in rural Honduras found acceptance of intimate partner violence tends to cluster in households, and acceptance of intimate partner violence in social networks was related to prevalence [78]. A social network study on the use of antenatal care among rural new mothers in Nepal showed that mothers-in-law generally had a negative influence on mothers’ use of antenatal care. The reasons they cited were few mothers-in-law had used antenatal care themselves, and so they thought it was unnecessary; low reading literacy and education among mothers-in-law; visits to the doctor took mothers away from their household responsibilities; and power dynamics within the household [22]. Mother-in-law perspectives on antenatal care predicted its use. These findings suggest focusing on health literacy within the mother’s core network has the potential to affect not only the mother’s health, but that of her child and her family, and that a nuanced view of relationships and relationship dynamics are needed to fully understand social networks effects. 

Within our sample, mothers or mothers-in-law and also men, primarily romantic partners, were frequently nominated by the mothers as health discussion partners. We noticed that, in general, male alters had smaller networks of health discussion partners than female alters. While all the mothers reported at least one person they would talk to, some men reported that there was no one in their social network they would ask health questions. Men in the study also reported low health literacy more frequently than the focal mothers. However, they were more likely to participate in the study, perhaps because they lived with the mother, providing a clearer bridge between them and the researchers. Men are also less likely to go to the doctor, especially when they are of working age [79,80]. This suggests that men in the home and their health literacy networks may be a feasible and important aspect of future health literacy network studies. 

Interventions focused on social networks are culturally relevant and a community asset that can be leveraged to improve health in racial/ethnic minority populations [81,82]. This study provided critical lessons on recruitment to inform future research and practice. 

### 4.3. Limitations and Future Studies

The study had some limitations. Due to the sample size and pilot nature of this work, many of the analyses were descriptive in nature. Our data are cross-sectional and these relationships may shift over time or around different health topics. Someone may feel more comfortable talking about children’s health with their mother or mother-in-law, but not mental health or sexual health. A network member may be crucial for one aspect of health, such as emotional support, but have low health literacy or misinformation about certain health topics. Additionally, social networks may change over time, especially related to major life events and changes such a having a child [24]. 

This work could also grow with more health measures to identify associations between network literacy and patient empowerment and engagement. It would be interesting to add a measure on shared decision making to determine whether women with higher health literacy, or alters who had higher health literacy, either felt more prepared or had a higher desire to engage in shared decision making. Shared decision making is critical to patient empowerment and engagement [83], but without sufficient information, (or people in their life who can help navigate the process) people tend to take a more passive role in health care encounters [84]. 

Other factors we were not able to consider in this pilot study that could be important to social network use and value include location (urban vs. rural), linguistic preferences, and immigration status. Our focus was on egocentric networks, not sociocentric networks, which would be of great value for fully understanding health literacy in a community, including information about the potential roles of centrality and density [20,23,24]. 

We focused recruitment on Native Hawaiian and Filipino communities. While these particular racial/ethnic groups may not be relevant to addressing health disparities in all communities, the focus on understanding the particular patterns of need and health literacy related practice in distinct community and cultural contexts is an important universal goal. Further, we note that the exact proportions of these racial/ethnic groups in Hawai‘i, a racial/ethnic minority majority state with a large mixed-race population [85], depends heavily on the way in which race/ethnicity is measured [86]. This complexity was, in fact, seen in our study population from the two mothers who self-identified as Puerto Rican mixed race during the interview though they were identified through the home visiting program as meeting our recruitment goals of being Native Hawaiian or Filipino. 

Our study focused on mothers. Other demographic groups may have different characteristics. For instance, social networks of older adults with low health literacy, in particular, may constitute people with more similar social and educational background [87]. This study took place in a very specific group of mothers in a very specific place. Furthermore, the egos in our study were all mothers availing themselves of needed social services. More isolated mothers, especially those with lower health literacy, may be less likely to participate home visiting or in a social network study. Nevertheless, we believe that this pilot research can provide insights not just to our community but to others. Mothers who participated in the home visiting program might have a higher level of health literacy than those who did not participate in the program. In future work, it would be interesting to determine whether health literacy improved from home visiting program participation. 

Finally, we used an egocentric perspective. Egocentric analyses also have strengths as well as weaknesses. Some strengths include the ability to quantitatively represent the importance of social relationships with flexible sampling frames and data collection strategies, to focus on individual-level outcomes, and to observe social similarities between people who influence each other’s decisions. Some limitations of egocentric analyses include a lack of reproducibility in findings and lack of focus on group-level outcomes.

## 5. Conclusions

Contextualizing health literacy within social networks can illuminate how individuals truly obtain, process, and understand basic health information and services to make appropriate health decisions. This study fills in key knowledge gaps concerning the topic of health literacy capacity within a health care decision network. Network health literacy can be conceptualized not just as a mediator or a moderator of individual health literacy, but as an empirical entity of its own that can be measured, tested, and changed. For some, it is the primary way in which health information is accumulated, evaluated, and used [8]. This conceptualization should be explored in future social network analysis, and this pilot study can provide some practical guidance around this. Our findings provide important evidence on the size and composition of health information networks of vulnerable new mothers to reduce health disparities and improve maternal and child health. We also provide practical information to inform future studies of health information networks. This research can inform future studies to meet the empirical research of ecological influences in health literacy and provides useful evidence on natural behavioral patterns and preferences to improve health and address health disparities.

## Figures and Tables

**Figure 1 ijerph-17-02356-f001:**
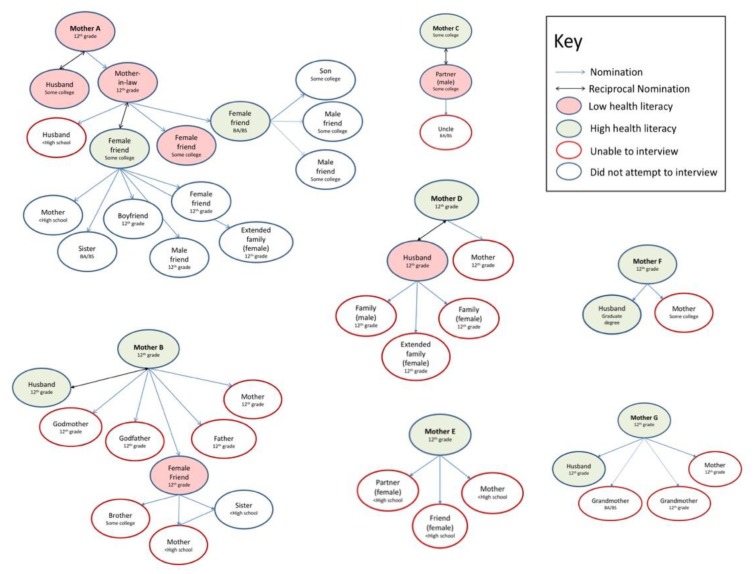
Egocentric health information structure of the seven focal moms (“egos”). Those we were able to interview have self-reported health literacy filled in. Self-reported low health literacy (LHL) is indicated by pink fill and high/adequate by grey. Those we tried but could not interview are outlined in red. Those who were not able to be interviewed according to the study protocol are outlined in blue.

**Table 1 ijerph-17-02356-t001:** Education and gender of all individuals in the networks.

	Mothers	Interviewed Alters (Levels 1 and 2)	Non-Interviewed Alters (Levels 1 and 2)
Participants (N = 31)	7	11	20
Percent Interviewed	100%	36%	
Education	n (%)	n (%)	n (%)
Less than HS	0 (0)	1 (9.1)	5 (25.0)
High School Graduate	7 (100)	8 (72.7)	11 (55.0)
Some College or Higher	0 (0)	2 (18.2)	4 (20.0)
Gender			
Female	7 (100)	6 (54.6)	13 (65)
Male	0 (0)	5 (45.5)	7 (35)

**Table 2 ijerph-17-02356-t002:** Health literacy and ethnicity of all individuals in the networks (Numbers sum to <100% due to rounding.)

	Mothers	Interviewed Alters (Levels 1 and 2)
Participants	7	11
	n (%)	n (%)
Self-Reported Health Literacy (“How confident do you feel filling out medical forms?”)		
Low (Not at all, A little bit, or Somewhat)	1 (14)	6 (55)
Adequate (Quite a bit or Extremely)	6 (86)	5 (45)
Ethnicity		
Native Hawaiian	4 (57)	3 (27)
Filipino	1 (14)	3 (27)
Puerto Rican Mix	2 (28)	0
Samoan/Tongan	0	1 (9)
Japanese	0	1 (9)
Other Pacific Islander	0	3 (27)
	Mean (SD)	Mean (SD)
Age	27 (3)	31 (9)

**Table 3 ijerph-17-02356-t003:** Education by Health Literacy among Individuals Interviewed.

Health Literacy Self-Reported	Education Self-Reported
Low	Grades 12 or GED
Adequate	Grades 12 or GED
Adequate	Grades 12 or GED
Adequate	Grades 12 or GED
Adequate	Grades 12 or GED
Adequate	Grades 12 or GED
Adequate	Grades 12 or GED
Low	College 1 to 3 yrs
Low	College 1 to 3 yrs
Adequate	College 1 to 3 yrs
Low	College 1 to 3 yrs
Adequate	Bachelor’s Degree
Low	Grades 12 or GED
Low	Grades 12 or GED
Adequate	Graduate degree
Adequate	Grades 12 or GED
Low	Grades 12 or GED
Adequate	Grades 9 thru 11

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
