# Peer review of "Considering Health Literacy, Health Decision Making, and Health Communication in the Social Networks of Vulnerable New Mothers in Hawai‘i: A Pilot Feasibility Study"

_ijerph, 2020, doi:10.3390/ijerph17072356_

Round 1
Reviewer 1 Report
The paper is a pilot study to examine the feasibility of interviews for their future health literacy study to inform future studies on the topic and describe low-income minority mother’s health literacy in the context of their social networks. It was an interesting pilot study. Overall the study is clear and well-written but there are few areas that need to be improved for publication.
- The authors stated the first goal of the study is to assess the feasibility of interviews for their future health literacy study to inform future studies on the topic, including strategies that were successful or unsuccessful. However, there was no discussion about the strategies in the results and discussions sections. Thus, the authors need to address these.
- The information about the interview instrument is very limited. The authors mentioned that they measured self-reported health literacy and asked basic demographics. However, from the results section, there seem to be other questions asked in the interviews. Thus, more information on the interview instrument needs to be provided in the text (e.g., other measures in the interviews, interview types - structure or semi-structured interviews).
- One of the study limitations is the source that the authors used for recruiting sample of mothers. The authors selected low-income mothers (egos) who “participate in a voluntary, evidence-based Home Visiting Program.” Those mothers who participated in the program might have a higher level of health literacy than those who did not in the program because the program’s “aim is to promote maternal and child health…”. It can be an explanation that only one mother had low health literacy in this study in the results.
- There are few typos.
Author Response
March 21, 2020
We appreciate the thoughtful consideration of this paper by the editors and reviewers and the opportunity to revise based on the feedback. The paper has been updated as detailed below. Changes to the manuscript are highlighted in blue in the tracked change document. Overall, we believe the paper is much improved and sincerely appreciate the time and wisdom of the reviewers in providing such useful and insightful feedback.
|
Yes |
Can be improved |
Must be improved |
Not applicable |
|
|
Does the introduction provide sufficient background and include all relevant references? |
(x) |
( ) |
( ) |
( ) |
|
Is the research design appropriate? |
(x) |
( ) |
( ) |
( ) |
|
Are the methods adequately described? |
( ) |
(x) |
( ) |
( ) |
|
Are the results clearly presented? |
(x) |
( ) |
( ) |
( ) |
|
Are the conclusions supported by the results? |
( ) |
( ) |
( ) |
( ) |
Comments and Suggestions for Authors
The paper is a pilot study to examine the feasibility of interviews for their future health literacy study to inform future studies on the topic and describe low-income minority mother’s health literacy in the context of their social networks. It was an interesting pilot study. Overall the study is clear and well-written but there are few areas that need to be improved for publication.
Thank you. We have revised towards this goal as detailed below.
- The authors stated the first goal of the study is to assess the feasibility of interviews for their future health literacy study to inform future studies on the topic, including strategies that were successful or unsuccessful. However, there was no discussion about the strategies in the results and discussions sections. Thus, the authors need to address these.
This is a great point. We have added a discussion of the strategies that were successful and unsuccessful in the results and discussions sections and implications for future studies on the topic. This can be found in section 4.1 in the discussion.
- The information about the interview instrument is very limited. The authors mentioned that they measured self-reported health literacy and asked basic demographics. However, from the results section, there seem to be other questions asked in the interviews. Thus, more information on the interview instrument needs to be provided in the text (e.g., other measures in the interviews, interview types - structure or semi-structured interviews).
Thank you! We have provided more information about the interview instrument in the methods section 2.2 measures.
- One of the study limitations is the source that the authors used for recruiting sample of mothers. The authors selected low-income mothers (egos) who “participate in a voluntary, evidence-based Home Visiting Program.” Those mothers who participated in the program might have a higher level of health literacy than those who did not in the program because the program’s “aim is to promote maternal and child health…”. It can be an explanation that only one mother had low health literacy in this study in the results.
We have added this good point to the study limitations section (starting at line 550).
- There are few typos.
We have resolved the typos.

Reviewer 2 Report
This paper deals with an interesting topic.
In the following please find my comments and suggestions:
- Introduction
A comprehensive literature review was carried out, but the conceptualization of vulnerability is missing. This also applies for the interrelations between vulnerability, health communication and health decision making of ethnic minorities.
The selection criteria for focusing on Native Hawaiians and Filipinos are not sufficiently described and it remains unclear why cultural relevance is relevant.
As we can find on https://www.to-hawaii.com/ethnicity.php that 10 % of the Hawaiian population is Native Hawaiians and other Pacific islanders, another 9 % are Hispanic. So why didn’t the authors of the study decide for the latter or Black or African American (who are about 2 %)?
The title of Chapter 1.2. “Study focus” is a bit misleading, because in this chapter deals with methodological considerations. Larger parts of the content of this chapter – especially related to SNA – should be summed up in a separate chapter. A presentation of the strengths and weaknesses of the “egocentric” SNA is missing.
The aim of the paper is not clearly described, research questions are missing: Is this a methodological-oriented paper or a research outcome-centered paper?
- Materials and methods
2.1. For this is a paper which presents results of a qualitative study, the description of sampling is too superficial and lacks in-depth information: What was the study or rather research context? Why did you choose “snowballing” as a sampling method and how did you carried it out? How long did it take to find the first test persons or rather interviewees?
There is no information provided on the considerations related to the creation of the health-related questions (did you only ask three questions?) and the duration of the interviews.
2.2. Measures are described too briefly. For readers who are not familiar with the Single Item Health Literacy Screener it is challenging to follow the argumentation and the way it is presented in this paper.
- Results
Figure 1 contains complex information and thus, needs a better graphical resolution and description in the results chapter. What are the key messages of Figure 1?
Table 2 relates to information on a small number of network members. Thus, the specification of percentage values is overdone.
Research findings do not relate to the issue of “vulnerability”.
- Discussion
In this chapter the relevance of vulnerability is addressed again, but cannot be interlinked to the research findings.
This is also true for many other theoretical and conceptual considerations which are touched upon in the discussion chapter. Due to the qualitative character of this pilot study and the quality of the empirical results this leads to a superficial discussion and lacks context to the vulnerability of ethnic minorities, their coping strategies and the compensating potential of their social networks.
The issue of incentivizing is addressed in order to be able to increase the readiness of more people in SNAs. Suggestions on how to do so are missing.
- Conclusions
The conclusions are too broad and superficial and lack context to vulnerability and ethnic minorities.

Author Response
March 21, 2020
We appreciate the thoughtful consideration of this paper by the editors and reviewers and the opportunity to revise based on the feedback. The paper has been updated as detailed below. Changes to the manuscript are highlighted in blue in the tracked change document. Overall, we believe the paper is much improved and sincerely appreciate the time and wisdom of the reviewers in providing such useful and insightful feedback.
This paper deals with an interesting topic. In the following please find my comments and suggestions:
- Introduction
A comprehensive literature review was carried out, but the conceptualization of vulnerability is missing. This also applies for the interrelations between vulnerability, health communication and health decision making of ethnic minorities.
Thank you for this excellent point. We have clarified vulnerability with this population, including the links across vulnerability, health communication and health decision making of ethnic minorities generally and specifically related to Native Hawaiian and Filipino populations in our study context. This can be found in the introduction (introducing why we focus on these groups, starting on line 93) as well as in the discussion section (considering the insights from our study, starting on line 367), and also scattered throughout.
The selection criteria for focusing on Native Hawaiians and Filipinos are not sufficiently described and it remains unclear why cultural relevance is relevant.
We have described this in more detail in the introduction, starting on line 104.
As we can find on https://www.to-hawaii.com/ethnicity.php that 10 % of the Hawaiian population is Native Hawaiians and other Pacific islanders, another 9 % are Hispanic. So why didn’t the authors of the study decide for the latter or Black or African American (who are about 2 %)?
Related to the point above, we have added more context for our choices around this issue in the introduction. Specifically, to the question of population proportion we add more detail. Hawaii has the highest percentage of multiracial Americans and is a notably diverse state. Measuring race/ethnicity in our multi-racial context is complex and, depending on how race/ethnicity is calculated, Native Hawaiians account for ~20-25% of the state population. The census data noted here is typically considered a deep undercount as mixed-race individuals are not fully accounted for who may identify as Native Hawaiians. When Census data accounts for this, the proportions of Native Hawaiians are much higher. We have added some detail and references about this issue in the discussion and limitation section, see especially the information starting on line 532.
The title of Chapter 1.2. “Study focus” is a bit misleading, because in this chapter deals with methodological considerations. Larger parts of the content of this chapter – especially related to SNA – should be summed up in a separate chapter. A presentation of the strengths and weaknesses of the “egocentric” SNA is missing.
We have revised the headings, reorganized the sections, and added a presentation of the strengths and weaknesses of the egocentric SNA in the limitations section (starting on line 553).
The aim of the paper is not clearly described, research questions are missing: Is this a methodological-oriented paper or a research outcome-centered paper?
We have added a clearer description of the paper goals, which can be found starting on line 159. We consider it a pilot feasibility study as it is (1) testing and describing methods towards feasibility and (2) providing preliminary data from a small sample related to research outcomes. We also consider it a landscaping paper to build future work/papers that could present clear study methods for a larger project (build from what we have learned from this pilot, feasibility study) and would result in outcomes from that study.
- Materials and methods
2.1. For this is a paper which presents results of a qualitative study, the description of sampling is too superficial and lacks in-depth information: What was the study or rather research context? Why did you choose “snowballing” as a sampling method and how did you carried it out? How long did it take to find the first test persons or rather interviewees?
We have added more information to explain the study design and timing to the materials and methods section.
There is no information provided on the considerations related to the creation of the health-related questions (did you only ask three questions?) and the duration of the interviews.
We have added more information to explain the questions asked and the interview duration to the materials and methods section.
2.2. Measures are described too briefly. For readers who are not familiar with the Single Item Health Literacy Screener it is challenging to follow the argumentation and the way it is presented in this paper.
We better explain the Single Item Health Literacy Screener and our other measures in the methods section.
- Results
Figure 1 contains complex information and thus, needs a better graphical resolution and description in the results chapter. What are the key messages of Figure 1?
We have provided more information about the messages of Figure 1 in the results, starting on line 236, and have improved the figures so they are easier to follow.
Table 2 relates to information on a small number of network members. Thus, the specification of percentage values is overdone.
We have removed the decimal.
Research findings do not relate to the issue of “vulnerability”.
This is a good point. We now tie this together more clearly, primarily in the framing of the introduction section and a consideration in the discussion section that links with the results presented here as mentioned above.
- Discussion
In this chapter the relevance of vulnerability is addressed again, but cannot be interlinked to the research findings.
We now included more detail on this important issue in the discussion.
This is also true for many other theoretical and conceptual considerations which are touched upon in the discussion chapter. Due to the qualitative character of this pilot study and the quality of the empirical results this leads to a superficial discussion and lacks context to the vulnerability of ethnic minorities, their coping strategies and the compensating potential of their social networks.
This is helpful as this is a key interest of this work. We have added more information about the context to the vulnerability of ethnic minorities, their coping strategies and the compensating potential of their social networks related to our study goals and results.
The issue of incentivizing is addressed in order to be able to increase the readiness of more people in SNAs. Suggestions on how to do so are missing.
We have added more detail on this, especially in the strategies section found starting on line 402
- Conclusions
The conclusions are too broad and superficial and lack context to vulnerability and ethnic minorities.
We have added context and details to resolve this generally and in our specific populations of interest.
Thank you again for your time and consideration!

Round 2
Reviewer 2 Report
Dear Editors,
The authors have taken into account all of the suggestions for improvement. Now the paper is of a very good quality and in my opinion ready for publication.
Kind regards,
Tatjana Fischer